# Should Microhematuria Be Incorporated into the 2023 Duke-International Society for Cardiovascular Infectious Diseases Minor Immunological Criteria?

**DOI:** 10.3390/antibiotics14070687

**Published:** 2025-07-07

**Authors:** Jean Regina, Louis Stavart, Benoit Guery, Georgios Tzimas, Pierre Monney, Lars Niclauss, Matthias Kirsch, Dela Golshayan, Matthaios Papadimitriou-Olivgeris

**Affiliations:** 1Department of Internal Medicine, Lausanne University Hospital and University of Lausanne, 1011 Lausanne, Switzerland; jean.regina@chuv.ch; 2Transplantation Center, Lausanne University Hospital and University of Lausanne, 1011 Lausanne, Switzerland; louis.stavart@chuv.ch (L.S.); dela.golshayan@chuv.ch (D.G.); 3Infectious Diseases Service, Lausanne University Hospital and University of Lausanne, 1011 Lausanne, Switzerland; benoit.guery@chuv.ch; 4Department of Cardiology, Lausanne University Hospital and University of Lausanne, 1011 Lausanne, Switzerland; georgios.tzimas@chuv.ch (G.T.); pierre.monney@chuv.ch (P.M.); 5Department of Cardiac Surgery, Lausanne University Hospital and University of Lausanne, 1011 Lausanne, Switzerland; lars.niclauss@chuv.ch (L.N.); matthias.kirsch@chuv.ch (M.K.); 6Infectious Diseases Service, Institut Central des Hôpitaux, Hospital of Valais, 1951 Sion, Switzerland

**Keywords:** infective endocarditis, microhematuria, urinalysis, acute kidney injury, Duke criteria, embolic events, bacteremia, sepsis

## Abstract

**Background/Objectives**: Microhematuria is common in patients with infective endocarditis (IE). The present study aims to assess whether the addition of microhematuria in the 2023 Duke-International Society for Cardiovascular Infectious Diseases (ISCVID) minor immunological criteria could enhance its diagnostic performance. **Methods**: This retrospective study was conducted at the Lausanne University Hospital, Switzerland (2014–2024). All patients with suspected IE and urinalysis within 24 h from presentation were included. The Endocarditis Team classified episodes as IE or non-IE. Microhematuria was defined as >5 red blood cells per high power field (HPF). **Results**: Among 801 episodes with suspected IE, 263 (33%) were diagnosed with IE. Microhematuria (>5/HPF) was present in 462 (58%) episodes, with no difference between episodes with and without confirmed IE (61% versus 56%; *p* = 0.223). Based on the 2023 ISCVID-Duke, minor immunological criteria were present in 42 episodes (5%). By adding microhematuria, 473 (59%) episodes met the minor immunological criteria. Sensitivity of the clinical criteria of the 2023 ISCVID-Duke version without and with hematuria was calculated at 75% (69–80%) and 86% (81–90%), respectively. Specificity was at 52% (48–57%) and 40% (36–45%), respectively. Among episodes with suspected IE, microhematuria was associated with female sex, enterococcal bacteremia, sepsis or septic shock, acute kidney injury, non-cerebral embolic events, and bone and joint infection. **Conclusions:** Microhematuria was frequent among patients with suspected IE, but it was not associated with the diagnosis of IE. The addition of microhematuria in the 2023 ISCVID-Duke minor immunological criteria did not enhance the overall performance of the criteria.

## 1. Introduction

Infective endocarditis (IE) is frequently associated with abnormal urinalysis, most commonly revealing proteinuria and microscopic hematuria [1,2,3,4,5]. Previous studies among patients with suspected IE demonstrated a higher frequency of microhematuria in those ultimately diagnosed with IE, suggesting its potential utility as a diagnostic marker [1,4].

Previous investigations have explored whether including microhematuria among the Duke criteria’s minor immunological phenomena affects its diagnostic accuracy [1,2,3]. In one series of 118 pathologically confirmed IE episodes, incorporating microhematuria (threshold not specified) increased the criteria’s sensitivity. In another cohort of 285 patients with suspected IE, defining microhematuria as >17 red blood cells per high-power field (HPF) led to 11% of episodes being reclassified from possible to definite IE. Similarly, in an evaluation of 163 suspected-IE episodes, inclusion of microhematuria (threshold not specified) also improved the sensitivity of the Duke criteria. Although, the authors of these studies proposed to incorporate microhematuria as a minor immunological criterion within the Duke diagnostic framework for IE, the small number of patients without IE in these studies precluded a robust assessment of specificity [1,2,3].

In 2023, both the International Society for Cardiovascular Infectious Diseases (ISCVID) and the European Society of Cardiology released updated Duke criteria. In both versions, only glomerulonephritis is recognized under the category of minor immunological phenomena [6,7]. Specifically, the 2023 ISCVID-Duke criteria defined glomerulonephritis as unexplained presence of either acute kidney injury or acute on chronic kidney injury plus two of the following findings: hematuria, proteinuria, cellular casts on inspection of urinary sediment, or serologic perturbations (hypocomplementemia, cryoglobulinemia, and/or presence of circulating immune complexes) [6].

This discrepancy highlights a gap in the current diagnostic framework, warranting further investigation into the role of microhematuria in IE diagnosis [6,7,8]. To address this gap, we aimed to assess the diagnostic performance of the 2023 ISCVID-Duke criteria by incorporating microhematuria as a minor immunological criterion. Additionally, we sought to identify factors associated with microhematuria in patients with suspected IE and those ultimately diagnosed with IE.

## 2. Results

Among 1855 episodes with suspected IE, 801 (43%) were included (Figure 1). The most common diagnosis was IE (263; 33%), followed by bone and joint infection (136; 17%). Non-infectious etiologies were diagnosed in 79 (10%) episodes. Transthoracic, transesophageal echocardiography (TTE, TEE), ^18^F-fluorodeoxyglucose positron emission tomography/computed tomography (^18^F-FDG PET/CT), and cardiac CT were performed for 762 (95%), 392 (49%), 192 (24%), and 38 (5%) episodes, respectively.

Microhematuria (>5/HPF) was present in 462 (58%) episodes, with no difference between IE and non-IE episodes (61% versus 56%; *p* = 0.223). Table 1 summarizes the clinical and laboratory parameters of episodes with and without IE.

According to the 2023 ISCVID-Duke criteria, the minor immunological criteria was met in 42 (5%) episodes (Table 2), with glomerulonephritis identified in 11/801 (1%) of episodes. When microhematuria > 5/HPF was incorporated as a minor immunological criterion, 473 (59%) episodes met this criterion. Using a higher threshold of >17/HPF, microhematuria was present in 363 (45%) episodes, leading to 378 (47%) episodes fulfilling the minor immunological criteria.

Table 3 presents the diagnostic performance of different versions of the 2023 ISCVID-Duke clinical criteria before and after incorporating microhematuria. Sensitivity of the original 2023 ISCVID-Duke criteria (without microhematuria) was 75%, compared to 86% and 83% for the versions including microhematuria at >5/HPF and >17/HPF, respectively. Specificity was 52% for the original criteria, decreasing to 40% with microhematuria > 5/HPF and 43% with microhematuria > 17/HPF.

Table 4 compares episodes with and without microhematuria (>5/HPF) among patients with suspected IE. In multivariable logistic regression analysis (Appendix A), microhematuria was associated with female sex (aOR: 1.60, 95% CI 1.14–2.26), enterococcal bacteremia (2.79, 1.55–5.04), sepsis or septic shock (1.67, 1.20–2.31), non-cerebral embolic events (2.08, 1.32–3.26), AKI upon presentation (1.47, 1.06–2.05), and bone and joint infection (2.09, 1.35–3.24). However, IE itself was not significantly associated with microhematuria (0.92, 0.65–1.30).

The comparison of episodes with and without microhematuria (>5/HPF) among the 263 episodes with diagnosed IE is shown in Appendix A. The multivariable logistic regression analysis (Appendix A) showed that among IE episodes, microhematuria was associated with non-cerebral embolic events (aOR: 2.40, 95% CI 1.32–4.36), AKI upon presentation (2.83, 1.50–5.34), and IE not related to prosthetic valves (2.14, 1.14–4.04).

## 3. Discussion

In our cohort of patients with suspected IE, incorporating microhematuria into the minor immunological criteria did not improve the overall performance of the 2023 ISCVID-Duke criteria.

Microhematuria was present in the majority of IE episodes (61%); however, a similar proportion (54%) was observed in episodes where IE was initially suspected but ultimately excluded. The reported incidence of microhematuria in IE patients varies widely in the literature (19–67%) [1,2,3,4,5,9]. Three studies that included patients with suspected IE who were ultimately not diagnosed found a lower incidence of microhematuria compared to those with confirmed IE [1,3,4]. This led to the suggestion that microhematuria should be incorporated into the Duke immunological criteria for IE diagnosis [1,2,3]. However, these studies had notable limitations. They included relatively small cohorts (118–285 episodes) and underrepresented episodes of suspected IE that were ultimately rejected (0–11%). In contrast, our study is the largest to date, including 801 episodes, with episodes without IE accounting for 67% of the total. Additionally, previous analyses were based on earlier versions of the Duke criteria, which have since been shown to have lower sensitivity compared to the 2023 ISCVID version [10,11,12,13,14,15].

In our study, adding microhematuria (either defined as >5/KPF or >17/KPF) to the 2023 ISCVID-Duke minor immunological criteria increased sensitivity but reduced specificity. This is explained by the lack of a significant difference in microhematuria incidences between episodes with and without IE. Most renal lesions in IE are of non-immunological origin, including infarcts, acute interstitial nephritis, and acute tubular necrosis [16]. Therefore, microhematuria alone lacks sufficient discriminatory power to aid in IE diagnosis and should not be classified as an immunological phenomenon. Furthermore, applying the modified 2023 ISCVID-Duke criteria with microhematuria in routine practice may increase false-positive IE diagnoses, particularly in settings with limited access to advanced imaging (^18^F-FDG PET/CT or cardiac CT) that could help in excluding IE [17] and lead to unnecessarily prolonged antimicrobial therapy.

Microhematuria was associated with non-cerebral embolic events at presentation, as previously observed by Ghosh et al. [5]. The high incidence of embolic events in our study (51%) may explain the higher prevalence of microhematuria compared to the study by Palepu et al., where embolic events occurred in only 26% of episodes [4]. Additionally, Majumbar et al. found that nearly half (45%) of IE episodes had renal infarcts at autopsy [16]. Many patients may also experience renal microinfarcts that are too small to be detected by imaging modalities such as abdominal CT [18], yet these can still result in urine abnormalities.

Consistent with a previous study linking microhematuria and AKI in IE patients [5], our study also found an association between microhematuria and AKI, both in patients with suspected IE and in those with confirmed IE. This is not unexpected, as suspected IE patients may have other contributing factors for microhematuria, such as nephrotoxic drugs or sepsis [19], with the latter also being independently associated with microhematuria. Sepsis can cause tubular injury through multiple mechanisms, the most significant being microcirculatory dysfunction [20]. The association of microhematuria with renal embolic events, AKI, or sepsis through non-immunologic mechanisms further complicates its inclusion as a minor immunological criterion.

Our study has several limitations. It was conducted at a single university hospital where infectious disease specialists systematically evaluate all suspected IE episodes, and advanced imaging modalities, such as ^18^F-FDG PET/CT, cardiac CT for valvular and paravalvular assessment, and cerebral and thoracoabdominal imaging for embolic event detection, are routinely used [18,21]. This specialized setting may limit the generalizability of our findings. Second, urine testing was not universally performed within the first 24 h of presentation. However, to our knowledge, this is the largest study to date evaluating microhematuria in episodes of suspected IE [1,2,3,4,5,9]. Additionally, the use of an Endocarditis Team to assess cases may have introduced misclassifications. This approach was necessary due to the absence of a definitive gold standard for IE diagnosis, which relies on multidisciplinary and highly specialized evaluation. Furthermore, while we excluded patients with conditions that could impact urinalysis results, such as urinary tract infections and urinary catheterization, data on other potential confounders such as menstruation during urine collection, were not recorded; however, only 55 (7%) episodes occurred in female patients younger than 51 years, which is the median age of menopause onset.

## 4. Materials and Methods

This single-center study was conducted at Lausanne University Hospital, Switzerland, from January 2014 to June 2024 (2014–2017, retrospective cohort; 2018–2024, prospective cohort).

We included adult patients with suspected IE (blood cultures drawn and echocardiography performed specifically for IE search) along with written consent (prospective cohort) or absence of refusal to use their data (retrospective cohort). Exclusion criteria were absence of urinalysis within 24 h from presentation, urinary tract infection diagnosis, and urinary tract catheterization.

Demographic, clinical, imaging, microbiological, surgical, and pathological data were manually retrieved from patients’ electronic health records. All data were reviewed by an infectious disease consultant. In our institution, infectious disease consultation was mandatory for all patients with suspected IE.

Each episode was classified as IE or non-IE based on the evaluation of the institution’s Endocarditis Team (reference standard). The determination of the infection site was based on the assessment by the infectious disease consultant responsible for the case, taking into account clinical, radiological, microbiological, and operative findings. Sepsis and septic shock were defined based on the Sepsis-3 International Consensus [22]. Immunological phenomena were defined as the presence of positive rheumatoid factors, Osler nodes, Roth spots, or glomerulonephritis, as described by the 2023 ISCVID-Duke criteria [6]. Acute kidney injury (AKI) was defined based on the 2012 KDIGO guidelines [23], and chronic kidney disease was estimated as a glomerular filtration rate < 60 mL/min/1.73 m^2^. Microhematuria was defined as the presence of >5 red blood cells per HPF [24].

Each episode was classified as definite, possible, or rejected IE according to the 2023 ISCVID-Duke clinical criteria, applied both with and without the inclusion of microhematuria (>5/HFP) as a minor immunological criterion. For this evaluation, an additional threshold of >17/HPF for microhematuria, as proposed by van der Vaart et al. [1], was also applied.

SPSS version 26.0 (SPSS, Chicago, IL, USA) was used for data analyses. A Fisher exact test or *chi*-square test was used for categorical variables, and a Mann–Whitney *U* test was used for continuous variables. Sensitivity, specificity, positive and negative predictive values (PPV, NPV), and accuracy were calculated with 95% confidence intervals (CIs). Episodes with IE according to the reference standard (Endocarditis Team evaluation) who were classified as definite IE by the Duke criteria were considered true positives, while those classified as possible or rejected IE were considered false negatives. Among episodes without IE according to the reference standard, those classified as rejected IE by the Duke criteria were considered true negatives, whereas episodes categorized as possible or definite IE were treated as false positives. Variables with *p* < 0.1 in the bivariable analyses that did not contribute to multicollinearity, assessed through the variance inflation factor, were used in multivariable logistic regression analyses. Adjusted odds ratios (aORs) and 95% CIs were calculated, and *p* < 0.05 was considered statistically significant.

## 5. Conclusions

Microhematuria was frequently observed in patients with suspected IE but was not associated with an IE diagnosis. Consequently, adding microhematuria to the 2023 ISCVID-Duke minor immunological criteria did not enhance the overall performance of the criteria, as it increased sensitivity at the expense of specificity. Microhematuria was associated with AKI, sepsis, and non-cerebral embolic events. Future studies should investigate the potential role of microhematuria and other urinary abnormalities in the diagnosis of IE.

## Figures and Tables

**Figure 1 antibiotics-14-00687-f001:**
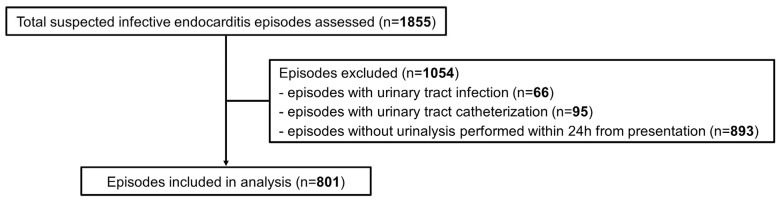
Flowchart of included patients.

**Table 1 antibiotics-14-00687-t001:** Comparison of episodes with and without infective endocarditis.

	No Infective Endocarditis (*n* = 538)	Infective Endocarditis (*n* = 263)	*p*
Demographics			
Male sex	378 (70)	208 (79)	0.008
Age (years)	69 (57–79)	66 (50–74)	0.005
Age > 60 years	370 (69)	163 (62)	0.067
Cardiac predisposing factors			
Intravenous drug use	21 (4)	38 (14)	<0.001
Congenital disease	13 (2)	38 (14)	<0.001
Prosthetic valve including transcatheter aortic valve replacement	50 (9)	85 (32)	<0.001
Prior endocarditis	11 (2)	34 (13)	<0.001
Moderate/severe valve regurgitation/stenosis	23 (4)	13 (5)	0.7117
CIED	47 (9)	40 (15)	0.008
Microbiological data			
Bacteremia/candidemia	404 (75)	247 (94)	<0.001
*S. aureus*	193 (36)	111 (42)	0.088
Coagulase-negative staphylococci	42 (8)	14 (5)	0.238
*Streptococcus* spp.	79 (15)	70 (27)	<0.001
*Enterococcus* spp.	36 (7)	32 (12)	0.010
Other Gram-positive	16 (3)	7 (3)	1.000
HACEK	3 (0.6)	3 (1)	0.400
Gram-negative other than HACEK	60 (11)	9 (3)	<0.001
*Candida* spp.	20 (4)	4 (2)	0.121
Microorganisms that occasionally or rarely cause IE isolated from at least three blood culture sets	15 (3)	14 (5)	0.105
New typical microorganism in the presence of intracardiac prosthetic material	80 (15)	20 (8)	0.003
Positive serology for *Coxiella burnetiid* or *Bartonella henselae*/*quintana*	1 (0.2)	1 (0.4)	0.549
Imaging data			
Positive echocardiography for vegetation, perforation, abscess, aneurysm, pseudoaneurysm, fistula	7 (1)	156 (59)	<0.001
Abnormal metabolic activity in ^18^F-FDG PET/CT	1 (0.2)	39 (15)	<0.001
Positive cardiac-CT for vegetation, perforation, abscess, aneurysm, pseudoaneurysm, fistula	1 (0.2)	20 (8)	<0.001
Significant new valvular regurgitation on echocardiography as compared to previous imaging	17 (3)	93 (35)	<0.001
Manifestations			
Fever (temperature > 38 °C)	439 (82)	232 (88)	0.019
Immunological phenomena ^a^	14 (3)	28 (11)	<0.001
Glomerulonephritis ^a^	2 (0.4)	9 (3)	0.001
Embolic events ^a^	65 (12)	159 (61)	<0.001
Hematogenous osteoarticular septic complications	46 (9)	45 (17)	0.001
Septic arthritis	24 (5)	26 (10)	0.005
Vertebral and non-vertebral osteomyelitis	31 (6)	23 (9)	0.133
Urinalysis results			
Red blood cells (×10^6^/L)	10 (0–80)	20 (0–80)	0.190
Microhematuria (red blood cells > 5/HPF)	302 (56)	160 (61)	0.223
Microhematuria (red blood cells > 17/HPF)	231 (43)	132 (50)	0.059
White blood cells (×10^6^/L)	1 (0–70)	0 (0–25)	0.018
Pyuria (white blood cells > 10/HPF)	233 (43)	96 (37)	0.067
Proteinuria (g/L)	0.25 (0–0.75)	0.25 (0–0.75)	0.650
Proteinuria (>0.3 g/L)	224 (42)	112 (43)	0.819
Renal function upon presentation			
Creatinine (μmol/L)	111 (76–169)	110 (81–167)	0.868
Acute kidney injury	199 (37)	110 (42)	0.190
Stage I	139 (70)	64 (58)	0.049
Stage II	35 (18)	20 (18)	
Stage III	25 (13)	26 (24)	
Data on surgery/CIED-extraction/histopathology			
Valve surgery performed	10 (2)	97 (37)	<0.001
CIED-extraction (among 87 patients with CIED)	4 (9)	18 (45)	<0.001
Autopsy performed	4 (0.7)	8 (3)	0.024
Histopathology compatible for IE	0 (0)	50 (19)	<0.001
Positive culture of vegetation, abscess	0 (0)	37 (14)	<0.001
Positive nucleic acid-based tests	0 (0)	13 (5)	<0.001
Macroscopic evidence of IE by inspection (surgery/autopsy)	0 (0)	66 (25)	<0.001

Data are depicted as number (%) or median (interquartile range); ^a^: as described by the 2023 International Society of Cardiovascular Infectious Diseases-Duke criteria; HACEK: *Haemophilus* spp., *Aggregatibacter* spp., *Cardiobacterium hominis*, *Eikenella corrodens*, *Kingella kingae*; HPF: high power field.

**Table 2 antibiotics-14-00687-t002:** Classifications based on the 2023 ISCVID version of the Duke clinical criteria before and after addition of microhematuria in the minor immunological criteria.

	No Infective Endocarditis (*n* = 538)	Infective Endocarditis (*n* = 263)
Duke major clinical criteria		
Major imaging criterion	23 (4)	194 (74)
Major surgery criterion	0 (0)	4 (2)
Major microbiological criterion	228 (42)	223 (89)
Duke minor clinical criteria		
Minor microbiological criterion	123 (19)	8 (5)
Minor predisposition criterion	145 (27)	184 (70)
Minor vascular criterion	65 (12)	159 (41)
Minor fever criterion	439 (82)	232 (88)
Minor immunological criterion (without microhematuria; original version)	14 (3)	28 (11)
Minor immunological criterion (with microhematuria > 5/HPF)	305 (57)	168 (64)
Minor immunological criterion (with microhematuria > 17/HPF)	235 (44)	143 (55)
Classification according to 2023 ISCVID-Duke clinical criteria without microhematuria (original version)		
Rejected	282 (52)	1 (0.4)
Possible	241 (45)	66 (25)
Definite	15 (3)	196 (75)
Classification according to 2023 ISCVID-Duke clinical criteria with microhematuria > 5/HPF		
Rejected	217 (40)	0 (0)
Possible	266 (49)	38 (14)
Definite	55 (10)	225 (86)
Classification according to 2023 ISCVID-Duke clinical criteria with microhematuria > 17/HPF		
Rejected	231 (43)	0 (0)
Possible	263 (49)	44 (17)
Definite	44 (8)	219 (83)

Data are depicted as number (percentage) or median (Q1–3); HPF: high power field; ISCVID: International Society of Cardiovascular Infectious Diseases.

**Table 3 antibiotics-14-00687-t003:** Performance of the different versions of the 2023 ISCVID-Duke clinical criteria before or after the addition of microhematuria in the minor immunological criteria.

	Sensitivity % (95% CI)	Specificity % (95% CI)	PPV % (95% CI)	NPV % (95% CI)	Accuracy % (95% CI)
Without microhematuria (original version)	75 (69–80)	52 (48–57)	43 (41–46)	81 (77–84)	60 (56–63)
With microhematuria > 5/HPF	86 (81–90)	40 (36–45)	41 (39–43)	85 (81–89)	55 (52–59)
With microhematuria > 17/HPF	83 (78–88)	43 (39–47)	42 (39–44)	84 (80–87)	56 (53–60)

CI: Confidence interval; HPF: high power field; ISCVID: International Society of Cardiovascular Infectious Disease; NPV: negative predictive value; PPV: positive predictive value.

**Table 4 antibiotics-14-00687-t004:** Comparison of episodes with suspected infective endocarditis with and without microhematuria upon presentation.

	Without Microhematuria (*n* = 339)	With Microhematuria (*n* = 462)	*p*
Demographics			
Male sex	259 (76)	327 (71)	0.090
Age (years)	66 (53–76)	70 (55–79)	<0.001
Age > 60 years	211 (62)	322 (70)	0.028
Comorbidities			
Diabetes mellitus	74 (22)	123 (27)	0.135
Obesity (body mass index ≥ 30 kg/m^2^)	67 (20)	107 (23)	0.261
Chronic kidney disease (eGFR < 60 mL/min/1.73 m^2^)	78 (23)	120 (26)	0.362
Malignancy (solid organ or haematologic)	77 (23)	93 (20)	0.383
Chronic obstructive pulmonary disease	41 (12)	47 (10)	0.424
Cirrhosis	28 (8)	42 (9)	0.706
Congestive heart failure	39 (12)	47 (10)	0.565
Manifestations upon presentation			
Fever (temperature > 38 °C)	276 (81)	395 (86)	0.146
Sepsis or septic shock	103 (30)	219 (47)	<0.001
Embolic events upon presentation ^a^	39 (19)	150 (25)	0.124
Cerebral embolic events	20 (10)	63 (11)	0.894
Non-cerebral embolic events	24 (12)	116 (19)	0.018
Renal function upon presentation			
Creatinine (μmol/L)	95 (69–133)	133 (87–190)	<0.001
Acute kidney injury	99 (29)	210 (46)	<0.001
Stage I	81 (82)	122 (58)	<0.001
Stage II	16 (16)	39 (18)	
Stage III	2 (2)	49 (23)	
Diagnosis			
Non-infectious diagnosis	48 (14)	31 (7)	0.001
Bacteremia/candidemia of unknown origin	22 (7)	44 (10)	0.152
Catheter-related	59 (17)	49 (11)	0.006
Low-respiratory tract infection	21 (6)	29 (6)	1.000
Abdominal infection	15 (4)	9 (2)	0.057
Skin and soft tissue infection	32 (9)	38 (8)	0.613
Bone and joint infections ^b^	37 (11)	99 (21)	<0.001
Septic arthritis	9 (3)	41 (9)	<0.001
Vertebral and non-vertebral osteomyelitis	16 (5)	38 (8)	0.063
Osteoarticular implant-associated infection	10 (3)	22 (5)	0.208
Infective endocarditis	103 (30)	160 (35)	0.223
Other infection	47 (14)	88 (19)	0.056
Bacteremia/candidemia	253 (75)	398 (86)	<0.001
*S. aureus*	112 (33)	192 (42)	0.015
Coagulase-negative staphylococci	27 (8)	29 (6)	0.401
*Streptococcus* spp.	73 (22)	76 (17)	0.081
*Enterococcus* spp.	18 (5)	50 (11)	0.007
Other Gram-positive	13 (4)	10 (2)	0.199
HACEK	1 (0.3)	5 (1)	0.410
Gram-negative other than HACEK	31 (9)	38 (8)	0.703
*Candida* spp.	7 (2)	17 (4)	0.213

Data are depicted as number (%) or median (interquartile range); ^a^: as described by the 2023 International Society of Cardiovascular Infectious Diseases-Duke criteria; ^b^: excluding chronic osteitis; eGFR: estimated Glomerular Filtration Rate; HACEK: *Haemophilus* spp., *Aggregatibacter* spp., *Cardiobacterium hominis*, *Eikenella corrodens*, *Kingella kingae.*

## Data Availability

The data that support the findings of this study are available from the corresponding author upon reasonable request.

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
