# Peer review of "Should Microhematuria Be Incorporated into the 2023 Duke-International Society for Cardiovascular Infectious Diseases Minor Immunological Criteria?"

_antibiotics, 2025, doi:10.3390/antibiotics14070687_

Round 1
Reviewer 1 Report
Comments and Suggestions for Authors
- Though the manuscript addressed clinical question, the central hypothesis is not clearer in the introduction. It is required to validate microhematuria as an independent diagnostic criterion or to evaluate the touch on sensitivity/specificity of updated Duke-ISCVID criteria.
- The manuscript uses two thresholds (>5/hpf and >17/hpf). Justify both cut-offs, and whether any ROC analysis was conducted to regulate optimal sensitivity/specificity trade-off.
- Given that including microhematuria raised sensitivity but reduced specificity, could you discuss the clinical consequences of potential overdiagnosis; particularly, in resource-limited areas in which misclassification could result in unnecessary antimicrobial therapy?
- The procedure of variable selection should be explicit for the multivariable logistic regression analysis. Were confounding and collinearity examined by more methods than the variance inflation factor? Were interactions explored?
- The analysis comprises 801 episodes. It is required to address whether the sample size was powered to detect differences in sensitivity/specificity.
- You refer to non-immunologic etiology of hematuria including renal infarcts. It is required to expand more on the concept that hematuria could be due to embolies or AKI and this complexing in terms of immunologic end point?
- Images Need More Accurate Captions and Axis Labels
- Use of figure 1 (flow diagram) would help but needs the box counted, inclusion/exclusion criteria and labels applied to arrows.
- Tables 1-4 should indicate P-values for significance with asterisks or bolding.
- Presentation of results submerged in numbers
- There is, however, a lot of redundancy of information between the text in Results and the numbers in tables. It is required to consider summarizing trends and highlighted key contrasts rather than restating percentages.
- Keep terminology consistent: “episodes,” “cases,” and “patients” are used interchangeably. Also, make sure you refer throughout to the “Duke criteria” as “2023 Duke-ISCVID criteria” where appropriate.
- Supplementary Tables 1-3 are cited in the main text without a brief summary of their content. A short summary should be added for better reader context.
- Some of the sentences are too complicated, or contort in texture. Examples:
“Urinalysis is often abnormal in patients with infective endocarditis...” → May need to be rewritten for flow.
“Cases were classified as definite, possible, or rejected...” → be more parallel and make it clear.
Author Response
Manuscript “Should Microhematuria Be Incorporated into the 2023 Duke-International Society for Cardiovascular Infectious Diseases minor immunological criterion?” We thank the Editor and all Reviewers for the in-depth review of our manuscript. We have addressed all comments in a point-by-point reply. We believe that the manuscript has been substantially improved.
Points raised by Reviewer 1
Point 1: Though the manuscript addressed clinical question, the central hypothesis is not clearer in the introduction. It is required to validate microhematuria as an independent diagnostic criterion or to evaluate the touch on sensitivity/specificity of updated Duke-ISCVID criteria.
Response: The central hypothesis is that incorporating microhematuria into the 2023 Duke-ISCVID criteria may alter its diagnostic performance, specifically sensitivity and specificity. We have revised the Introduction accordingly to clarify this point.
Point 2: The manuscript uses two thresholds (>5/hpf and >17/hpf). Justify both cut-offs, and whether any ROC analysis was conducted to regulate optimal sensitivity/specificity trade-off.
Response: The cut-off of microhematuria in the literature differs (>2, >3, >5). We decided to use the higher cut-off (>5/hpf) in order to reduce false‑positives and enhance specificity. We also used the >17/hpf to compare our results with the previous study from van der Vaart et al. We added a reference for the >5/hpf cut-off and we described the study of van der Vaart et al. in the Introduction.
Point 3: Given that including microhematuria raised sensitivity but reduced specificity, could you discuss the clinical consequences of potential overdiagnosis; particularly, in resource-limited areas in which misclassification could result in unnecessary antimicrobial therapy?
Response: A phrase is added to the third paragraph of Discussion.
Point 4: The procedure of variable selection should be explicit for the multivariable logistic regression analysis. Were confounding and collinearity examined by more methods than the variance inflation factor? Were interactions explored?
Response: As noted to the last paragraph of the Materials and Methods, variables with P<0.1 in the bivariable analyses that did not contribute to multicollinearity were used in multi-variable logistic regression analyses. Multicollinearity was assessed through variance inflation factor alone. No variable was highly correlated.
Point 5: The analysis comprises 801 episodes. It is required to address whether the sample size was powered to detect differences in sensitivity/specificity.
Response: Our dataset included 263 patients with IE and 538 without IE, which allowed us to assess changes in sensitivity and specificity of the Duke criteria with and without the addition of microhematuria. Based on the observed sensitivity improvement from 75% (95% CI: 69–80) to 86% (95% CI: 81–90) when using a microhematuria cutoff of >5 red blood cells per high-power field (hpf), and a specificity decrease from 52% (95% CI: 48–57) to 40% (95% CI: 36–45), we evaluated the power retrospectively. Assuming a two-sided alpha of 0.05 and a desired power of 90%, we calculated by the McNemar’s test that detecting a 10% absolute increase in sensitivity (from 75% to 85%) would require approximately 218 IE cases. Similarly, detecting a 10–12% absolute difference in specificity would require approximately 450 non-IE cases. Our study population exceeded both thresholds, with 263 IE cases and 538 non-IE cases, suggesting the sample was adequately powered to detect clinically relevant differences in diagnostic performance.
Point 6: You refer to non-immunologic etiology of hematuria including renal infarcts. It is required to expand more on the concept that hematuria could be due to embolies or AKI and this complexing in terms of immunologic end point?
Response: We expanded the discussion in the fifth paragraph to clarify that microhematuria can result from non-immunologic causes such as renal emboli or AKI, complicating its inclusion as an immunologic criterion.
Point 7: Images Need More Accurate Captions and Axis Labels
Point 8: Use of figure 1 (flow diagram) would help but needs the box counted, inclusion/exclusion criteria and labels applied to arrows.
Response: Figure 1 was revised accordingly.
Point 9: Tables 1-4 should indicate P-values for significance with asterisks or bolding.
Response: We agree that highlighting statistically significant p-values using asterisks or bold formatting could improve the readability of Tables 1–4. However, we are uncertain whether such formatting aligns with the journal’s style guidelines. We are happy to revise accordingly if the Editor advises so.
Point 10: Presentation of results submerged in numbers
Point 11: There is, however, a lot of redundancy of information between the text in Results and the numbers in tables. It is required to consider summarizing trends and highlighted key contrasts rather than restating percentages.
Response: We thank the reviewer for this important observation. We agree that the current Results section includes a substantial amount of numerical detail, some of which duplicates information in the tables. While some repetition is intentional to guide the reader through key findings, we acknowledge the benefit of streamlining the text. We have revised the Results. We hope this improves readability and clarity without omitting essential findings.
Point 12: Keep terminology consistent: “episodes,” “cases,” and “patients” are used interchangeably. Also, make sure you refer throughout to the “Duke criteria” as “2023 Duke-ISCVID criteria” where appropriate.
Response: We changed to terms to be consistent.
Point 13: Supplementary Tables 1-3 are cited in the main text without a brief summary of their content. A short summary should be added for better reader context.
Response: Supplementary Tables 1 and 3 depict the multivariate analyses. The significant results of these analyses appear in the Text. The Supplementary Table 2 depicts the comparison with and without microhematuria (>5/hpf) among the 263 episodes with diagnosed IE. To avoid redundancy between the text and the supplementary tables, we opted not to repeat non-essential details already presented in the tables.
Point 14: Some of the sentences are too complicated, or contort in texture. Examples:
“Urinalysis is often abnormal in patients with infective endocarditis...” → May need to be rewritten for flow.
“Cases were classified as definite, possible, or rejected...” → be more parallel and make it clear.
Response: We revised the identified sentences accordingly
Reviewer 2 Report
Comments and Suggestions for Authors
The authors conducted a retrospective and prospective cohort study of 801 episodes of suspected IE to evaluate whether adding microhematuria to the 2023 Duke ISCVID minor immunological criterion improves diagnostic accuracy. The conclusion that sensitivity increases but overall performance declines due to reduced specificity, is well supported by the data. The results are clearly presented, and the strengths and limitations are appropriately acknowledged. I have a few comments for the authors.
- Could the authors clarify the definition of “no infective endocarditis”? Does this refer to patients without bacterial infection or to those in whom IE was ultimately ruled out?
- Would it be appropriate to apply statistical tests to compare the diagnostic performance metrics shown in Table 3? If so, including this analysis would help reinforce the conclusions.
- The Methods section would benefit from more details on how sensitivity and specificity were calculated.
- A couple of minor typos should be corrected.
In Table 1, the p-value for “Autopsy performed” is incorrectly written as “0.0.24”
In Table 3, the confidence interval for “PPV% (95% CI)” in the “Without microhematuria” row is incomplete (“43 (41–4)”) and should be fixed.
Author Response
Manuscript “Should Microhematuria Be Incorporated into the 2023 Duke-International Society for Cardiovascular Infectious Diseases minor immunological criterion?” We thank the Editor and all Reviewers for the in-depth review of our manuscript. We have addressed all comments in a point-by-point reply. We believe that the manuscript has been substantially improved.
Points raised by Reviewer 2
Point 1: Could the authors clarify the definition of “no infective endocarditis”? Does this refer to patients without bacterial infection or to those in whom IE was ultimately ruled out?
Response: Each episode was classified as IE or non-IE based on the evaluation of the institution’s Endocarditis Team, which was the reference standard. We reformatted the definition in the Methods section.
Point 2: Would it be appropriate to apply statistical tests to compare the diagnostic performance metrics shown in Table 3? If so, including this analysis would help reinforce the conclusions.
Response: We agree that a statistical comparison of diagnostic classifications could provide additional insight. We therefore performed McNemar’s test to evaluate whether the inclusion of microhematuria significantly changed classification outcomes. The test confirmed a statistically significant difference between the original 2023 ISCVID Duke criteria and both modified versions incorporating microhematuria (P < 0.001). However, this test primarily demonstrates that the diagnostic classifications differ between the versions; it does not directly indicate which version is more accurate or clinically appropriate.
Point 3: The Methods section would benefit from more details on how sensitivity and specificity were calculated.
Response: We revised the Materials and Methods section (paragraphs 4-6) to better illustrate how sensitivity and specificity were calculated.
Point 4: A couple of minor typos should be corrected.
Response: We reviewed the manuscript and corrected it.
Point 5: In Table 1, the p-value for “Autopsy performed” is incorrectly written as “0.0.24”
Response: Value corrected.
Point 6: In Table 3, the confidence interval for “PPV% (95% CI)” in the “Without microhematuria” row is incomplete (“43 (41–4)”) and should be fixed.
Response: Value corrected.
Reviewer 3 Report
Comments and Suggestions for Authors
Reviewer evaluation
Keywords: infective endocarditis; microhematuria; urinalysis; retrospective study; Lausanne University Hospital; ISCVID.
I see that the introduction is very short, it is better to extend it by describing more the key words of this subject
Building on these observations, it was proposed ……….. of minor immunological phenomena.[6,7] Specifically, the 2023 International Society …….. and/or presence of circulating immune complexes).[6]
References write all journal names abbreviated and in italic
- Lamas, C.C.; Eykyn, S.J. Suggested modifications to the Duke criteria for the clinical diagnosis of native valve and prosthetic valve endocarditis: analysis of 118 pathologically proven cases. Clin infect dis. 1997, 25, 713-719, doi:10.1086/513765.
- Fowler, V.G.; Durack, D.T.; Selton-Suty, C.; Athan, E.; Bayer, A.S.; Chamis, A.L.; Dahl, A.; DiBernardo, L.; DuranteMangoni, E.; Duval, X.; et al. The 2023 Duke-ISCVID Criteria for Infective Endocarditis: Updating the Modified Duke Criteria. Clin infect dis. 2023, doi:10.1093/cid/ciad271
- van der Vaart, T.W.; Bossuyt, P.M.M.; Durack, D.T.; Baddour, L.M.; Bayer, A.S.; Durante-Mangoni, E.; Holland, T.L.; Karchmer, A.W.; Miro, J.M.; Moreillon, P.; et al. External Validation of the 2023 Duke - International Society for Cardiovascular Infectious Diseases Diagnostic Criteria for Infective Endocarditis. Clin infect dis. 2024, doi:10.1093/cid/ciae033.
- Lindberg, H.; Berge, A.; Jovanovic-Stjernqvist, M.; Hagstrand Aldman, M.; Krus, D.; Oberg, J.; Kahn, F.; Blackberg, A.; Sunnerhagen, T.; Rasmussen, M. Performance of the 2023 Duke-ISCVID diagnostic criteria for infective endocarditis in relation to the modified Duke criteria and to clinical management- reanalysis of retrospective bacteremia cohorts. Clin infect dis. 2024, doi:10.1093/cid/ciae040.
- Moisset, H.; Rio, J.; Benhard, J.; Arnoult, F.; Deconinck, L.; Grall, N.; Iung, B.; Lescure, X.; Rouzet, F.; Suc, G.; et al. Evaluation of the specificity of the 2023 Duke-International Society of Cardiovascular Infectious Diseases classification for infective endocarditis. Clin infect dis. 2024, doi:10.1093/cid/ciae034.
- Goehringer, F.; Lalloue, B.; Selton-Suty, C.; Alla, F.; Botelho-Nevers, E.; Chirouze, C.; Curlier, E.; El Hatimi, S.; Gagneux-Brunon, A.; le Moing, V.; et al. Compared Performance of the 2023 Duke-International Society for Cardiovascular Infectious Diseases, the 2000 Modified Duke, and the 2015 ESC Criteria for the Diagnosis of Infective Endocarditis in a French Multicenter Prospective Cohort. Clin infect dis. 2024, doi:10.1093/cid/ciae035. 16. Papadimitriou-Olivgeris, M.; Monney, P.; Frank, M.; Tzimas, G.; Tozzi, P.; Kirsch, M.; Van Hemelrijck, M.; Bauernschmitt, R.; Epprecht, J.; Guery, B.; et al. Evaluation of the 2023 Duke-ISCVID and 2023 Duke-ESC clinical criteria for the diagnosis of infective endocarditis in a multicenter cohort of patients with Staphylococcus aureus bacteremia. Clin infect dis. 2024, doi:10.1093/cid/ciae003. 17. Papadimitriou-Olivgeris, M.; Monney, P.; Frank, M.; Tzimas, G.; Tozzi, P.; Kirsch, M.; Van Hemelrijck, M.; Bauernschmitt, R.; Epprecht, J.; Guery, B.; et al. Evaluation of the 2023 Duke-ISCVID criteria in a multicenter cohort of patients with suspected infective endocarditis. Clin infect dis. 2024, doi:10.1093/cid/ciae039.
- Papadimitriou-Olivgeris, M.; Guery, B.; Ianculescu, N.; Dunet, V.; Messaoudi, Y.; Pistocchi, S.; Tozzi, P.; Kirsch, M.; Monney, P. Role of cerebral imaging on diagnosis and management in patients with suspected infective endocarditis. Clin infect dis. 2023, doi:10.1093/cid/ciad192.
Author Response
Manuscript “Should Microhematuria Be Incorporated into the 2023 Duke-International Society for Cardiovascular Infectious Diseases minor immunological criterion?” We thank the Editor and all Reviewers for the in-depth review of our manuscript. We have addressed all comments in a point-by-point reply. We believe that the manuscript has been substantially improved.
Points raised by Reviewer 3
Point 1: I see that the introduction is very short, it is better to extend it by describing more the key words of this subject
Building on these observations, it was proposed ……….. of minor immunological phenomena.[6,7] Specifically, the 2023 International Society …….. and/or presence of circulating immune complexes).[6]
Response: We have expanded the Introduction to better contextualize the study within the existing literature and to define the relevant key concepts, including infective endocarditis, microhematuria, and their diagnostic implications.
Point 2: References write all journal names abbreviated and in italic
Response: References are updated.